# Morally injurious events and post-traumatic embitterment disorder in UK health and social care professionals during COVID-19: a cross-sectional web survey

Chloe J Brennan ,[1] Michael T McKay,[2] Jon C Cole[1]

[1]Department of Psychology, University of Liverpool, Liverpool, UK
[2]Department of Psychology, Royal College of Surgeons in Ireland, Dublin, Ireland

**Correspondence to**
Chloe J Brennan;
hlcbrenn@liverpool.ac.uk

## ABSTRACT

**Objective** To estimate the prevalence and predictors of morally injurious events (MIEs) and post-traumatic embitterment disorder (PTED) in UK health and social care professionals during the COVID-19 pandemic.

**Design** Cross-sectional study.

**Setting** September–October 2020 in the UK. Online survey hosted on Qualtrics, and recruited through Prolific.

**Participants** 400 health and social care workers, aged 18 or above and living and working in the UK during the pandemic.

**Main outcome measures** MIEs were assessed using the Moral Injury Events Scale and PTED was assessed using the PTED self-rating scale. Potential predictors were measured using surveys of exposure to occupational stressors, optimism, self-esteem, resilient coping style, consideration of future consequences and personal belief in a just world.

**Results** 19% of participants displayed clinical levels of PTED, and 73% experienced at least one COVID-related MIE. Exposure to occupational stressors increased the risk of experiencing PTED and MIEs, whereas personal belief in a procedurally just world, which is the belief that they experienced fair processes, was a protective mechanism.

**Conclusions** MIEs and PTED are being experienced by UK health and social care professionals, particularly in those exposed to work-related stressors.

## INTRODUCTION

By January 2021, the rapid spread of the novel COVID-19 had claimed the lives of 100 000 people in the UK.[1] In the year prior to the pandemic, UK healthcare workers (HCWs) reported work-related stress at its highest level since 2014.[2] The early psychological impact of COVID-19 on HCWs is not equivocally experienced as globally, rates of depression (8.9%–50.4%), anxiety (14.5%–44.6%), post-traumatic stress disorder (PTSD; 8.3%–88.4%), stress (2.2%–93.8%) and burnout (21.8%–46.3%) in HCWs vary widely across studies.[3 4] Research has focused predominantly on HCWs, but preliminary

### Strengths and limitations of this study

⇒ This represents the first prevalence estimates of both morally injurious events (MIEs) and post-traumatic embitterment disorder (PTED) in UK health and social care workers (HSCWs) during COVID-19.

⇒ Findings are generalisable to HSCWs across diverse job roles rather than only 'front-line' healthcare staff.

⇒ To inform healthcare policy, validated scales identified demographic, occupational and personality traits associated with an increased risk of MIEs and PTED.

⇒ The cross-sectional design prevents analysis of causal relationships, and comparisons to prepandemic levels of PTED and MIEs are not possible.

⇒ Despite controlling for social desirability in analyses, survey data suffer from response and social desirability bias.

evidence into UK health and social care workers (HSCWs) during the pandemic indicates that nearly two-thirds met thresholds for clinically significant disorders (PTSD=22%, anxiety=47%, depression=47%).[5]

Health research in the wake of the COVID-19 pandemic has largely focused on symptoms of anxiety, depression, PTSD and stress. However, it is arguable that an equally pressing health-related concern is the way in which HSCWs feel they have been treated throughout the pandemic and the associated psychological impact. Individuals have risked their lives during shortages in staff, limited availability of personal protective equipment (PPE) and other resources, which can lead to feelings of injustice and increased pressure to make morally challenging decisions.[6] Doctors surveyed in December 2020 reported feeling unable to provide the right care at the right time, with around 40% experiencing feelings of anger/irritability.[7] These experiences have

the potential to violate healthcare staff's ethical beliefs, placing them at an increased risk of experiencing moral injury and post-traumatic embitterment disorder (PTED).

Witnessing or perpetrating events that violate one's moral values are called 'morally injurious events' (MIEs), and moral injury encapsulates symptoms such as guilt, shame and poor mental health outcomes associated with these events.[8] Traditionally, moral injury has been studied in military personnel after witnessing the horrors of war. However, there is growing recognition that moral injury can occur in civilian healthcare professionals, and some argue that morally injured healthcare staff are being inaccurately diagnosed with burnout.[9] PTED is a reactive disorder, triggered by an unjust event and is accompanied by intrusive thoughts, mood impairment, social withdrawal, hostility and feelings of helplessness.[10 11] Theoretically, PTED and moral injury are very similar. Their aetiologies are the same, as both develop following an event(s) that has breached an individual's core beliefs. After the event(s), symptoms span the emotional (ie, moral emotions of anger and shame), cognitive (ie, negative beliefs about oneself, others and the world) and social (ie, relational) domains.

The biggest distinction between PTED and moral injury is their approach to psychiatric diagnoses. Most PTED researchers agree that embitterment should be a separate psychiatric disorder in diagnostic manuals if it causes significant impairment to daily functioning for over 6 months.[10] In contrast, some moral injury researchers are hesitant to pathologize adaptive moral reactions to ethical issues and so moral injury is often considered a risk factor for other severe mental health issues.[6] Similarly, a useful conceptualisation of the PTED–moral injury relationship may be that exposure to MIEs can lead to PTED. Analysing this relationship can advance our understanding of moral trauma and provide a useful diagnosis for treatment resistant patients exhibiting these symptoms.

The British Medical Association has suggested that moral injury may become one of the most significant injuries for healthcare staff responding to the COVID-19 pandemic, and preliminary evidence in the USA[12] and China[13] appears to support this claim. Initial data from National Health Service (NHS) check, a cohort study of UK healthcare staff during the pandemic, indicates that MIEs are linked to increased rates of mental health symptoms.[14] However, there is a lack of studies in the UK investigating the prevalence of both moral injury and PTED in healthcare staff, despite their similarities. Additionally, despite an increased risk, not all staff members will develop PTED and/or moral injury as a result of their experiences during the pandemic. The literature suggests that individuals high in optimism, self-esteem, resilient coping and belief in a just world are at a reduced risk of experiencing PTED and/or moral injury,[8 15] and individuals who consider the future consequences of their behaviour have better health outcomes.[16] The current study reports on available baseline findings from a longitudinal study of UK HSCWs that began in September–October 2020.

The study aims to establish the prevalence of PTED and MIEs and identify potential risk and protective factors. Further waves of this study will investigate whether factors at baseline predict future PTED and/or MIEs, establish whether prevalence rates increase across time as the pandemic continues and investigate whether MIEs at baseline predict subsequent PTED.

## METHODS
### Design
This is a cross-sectional analysis of baseline data gathered after the first peak of COVID-19 in the UK (24 September to 6 October 2020).

### Participants and procedure
Participants were 400 HSCWs, aged 18–67, residing in the UK, and employed during the COVID-19 pandemic. Job roles were categorised in line with careers on the NHS website[17] (see online supplemental table 1) for self-reported demographics). Participants were recruited via Prolific, an online crowdsourcing platform and received £2.50 for completing the survey (in line with platform guidelines). A further wave of data will take place at a 12-month follow-up. Participants that volunteered were directed to an online survey hosted on Qualtrics where they provided informed consent, responded to psychological measures in a randomised order, were debriefed, and redirected back to Prolific. Prolific ensures no duplicate responders using IP address tracking. Due to participants potentially experiencing considerable levels of PTED and/or MIEs, appropriate support links and researchers' contact details were provided on the information and debrief sheets. The survey took approximately 15 min to complete. A power calculation conducted in G*Power found that 195 participants were required to find a small-to-moderate effect size of $f^2$=0.12 with power 0.80 and α=0.05. To account for attrition rates of >50% in longitudinal studies, 400 participants were recruited. Completeness was checked after survey completion, and incomplete responses were not analysed. This included participants that Prolific automatically eliminated because they took too long to complete the survey. Prior to study completion, 47 out of 447 participants dropped out, which is a response rate of 89.5%. To minimise non-response bias, study information was clearly communicated, and the survey was short.

### Patient and public involvement
No patients or the public were involved in the design or conduct of this study.

### Materials
Overall, 98 survey items were distributed across 11 pages, and items per page ranged from 1 to 19.

Participants self-reported their age, gender, ethnicity, professional role, years in current role and any current mental health diagnoses.

MIEs were assessed using the Moral Injury Events Scale (MIES),[18] a nine-item scale measuring exposure to military MIEs that violate moral beliefs. A higher score indicated more exposure to MIEs. The scale was adapted to reflect a health and social care sample in two ways. First, participants were asked to respond regarding their experiences as an HSCW professional since the outbreak of COVID-19. Second, three items assessing betrayal by 'leaders', 'fellow service members' and 'others outside of the US military' in the original scale were changed to 'superiors' 'fellow colleagues' and 'others outside of my work organisation'. Ratings were made on a 6-point Likert scale, ranging from 1 (strongly disagree) to 6 (strongly agree). A recent factor analysis found three reliable subscales of the MIES; (1) transgressions by others (transgressions-others; α=0.79), (2) transgressions by self (transgressions-self; α=0.94) and (3) betrayal (a=0.89).[19] The mean scores across items in each subscale were used in analyses. For prevalence estimates, we calculated the percentage of participants who 'slightly agreed', 'agreed' or 'strongly agreed' with each item of the MIES.

PTED was assessed using the PTED Self-Rating Scale,[11] which is a 19-item scale assessing embitterment following negative life events. Participants were asked to respond regarding working during the COVID-19 pandemic on a 5-point scale ranging from 0 (not true at all) to 4 (extremely true). The mean score across the 19 items was used in analyses relating to PTED. In line with recommendations,[11] a mean total score of ≥2 was considered to be indicative of clinically relevant levels of PTED.

To measure exposure to occupational stressors, a five-item scale was developed for this study. Participants indicated on a 4-point scale (1=not at all, 4=all the time) how often they experienced: (1) exposure to COVID-19 in their job; (2) a lack of PPE and/or clear training; (3) having to make difficult decisions regarding resource allocation; (4) an inability to provide adequate care or save lives; and (5) separation from, or fear for, loved ones due to working during the current pandemic. Items were summed, with higher scores reflecting higher exposure.

The Revised Life Orientation Test (LOT-R)[20] is a 10-item scale assessing generalised optimism. Responses (0=strongly disagree, 4=strongly agree) to four-filler items were removed. As a result of confirmatory factor analysis (see online supplemental materials), we employed a two-factor model suggested by Glaesmer et al,[21] where the sum of three items reflected higher optimism scores and the sum of three items reflected higher pessimism scores.

The Rosenberg Self-Esteem Scale[22] asked participants to indicate their agreement to 10 items using a 4-point Likert scale where 1=strongly disagree and 4=strongly agree. Five negatively worded items were recoded so that higher total scores reflected higher self-esteem.

The Brief Resilience Coping Scale[23] is a short scale measuring tendency to cope with stress adaptively. Four items were assessed on a 5-point Likert scale (1=does not describe at all, 4=describes me very well). Total scores range from 4 to 20, with high scores indicating resilient coping ability.

The Belief in a Just World Scale (BJW)[24] contains eight items measuring perceptions of fairness towards oneself (personal BJW) and eight items measuring perceptions of fairness towards others (general BJW). Personal BJW better predicts psychological well-being so only this scale was used. These eight items were rated on a 7-point Likert scale from 1 (strongly disagree) to 7 (strongly agree). Four items measure distributive justice and four items measure procedural justice.

The Consideration of Future Consequences-14 Scale (CFC-14)[25] uses the sum of seven items to assess consideration of future consequences- future (CFC-F; α=0.82) and the sum of seven items to assess consideration of immediate consequences-immediate (CFC-I; α=0.82). Responses are given on a 7-point Likert ranging from 1 (very unlike me) to 7 (very much like me) and no items are reverse scored.

Social desirability was measured using the 13-item Marlowe-Crowne Social Desirability Scale.[26] Participants indicated either 'true' (0) or 'false' (1) and five items were reverse coded so higher scores reflected more socially desirable responses. Social desirability was controlled for in hierarchical regression analyses so no participants were excluded based on high scores.

## Data analysis

Based on confirmatory factor analyses (CFA), we employed a three-factor solution for the MIES, and a two-factor solution for the LOT-R, the personal BJW and the CFC-14. All other scales used a unidimensional structure (see online supplemental materials). Prevalence estimates for PTED and MIES were calculated followed by Pearson's bivariate correlations to examine the relatedness of these constructs, and their relationship to predictor variables. Independent samples t-tests and a one-way analysis of variance (ANOVA) examined the impact of socio-demographic variables on PTED and MIES scores. Four hierarchical linear regressions were then conducted for each dependent variable (PTED, transgressions-others, transgressions-self and betrayal) to assess the unique effects of demographics and social desirability (step 1), occupational stressors (step 2) and personality characteristics (step 3). Pairwise deletion was used in univariate analyses, and listwise deletion in multivariate analyses. Bias-corrected and accelerated bootstrapping methods were applied to all analyses to deal with violations to normality, outliers and homoscedacity of variance, and all other assumptions were met. For all analyses, $p<0.05$ was deemed significant and we used Ferguson's[27] recommendations for minimum practical ($d \geq 0.41$, $r \geq 0.20$, $\omega^2 \geq 0.04$), medium ($d \geq 1.15$, $r \geq 0.50$, $\omega^2 \geq 0.25$) and large effect sizes ($d \geq 2.70$, $r \geq 0.80$, $\omega^2 \geq 0.64$). Cohen's $f^2$[28] was used to interpret the effect size of overall regressions, where $f^2 \geq 0.02$, $f^2 \geq 0.15$ and $f^2 \geq 0.35$ represented small, medium and large effect sizes, respectively. Recommendations by Perry et al[29] were used to interpret CFA where

**Table 1** PTED prevalence and frequency of endorsement of items on the MIES in UK health and social care workers

| Cut-off score | Raw frequency (N) | % (95% CI) |
|---|---|---|
| PTED Scale | | |
| 1.6 | 121 | 30.3% (25.8% to 35%) |
| 2 | 77 | 19.3% (15.5% to 23.5%) |
| 2.5 | 30 | 7.5% (5.1% to 10.5%) |
| MIES Scale | | |
| Any scale item | 289 | 72.3% (67.6% to 76.6%) |
| Transgressions by others subscale | 213 | 53.3% (48.2% to 58.2%) |
| Transgressions by self subscale | 131 | 32.8% (28.2% to 37.6%) |
| Betrayal subscale | 227 | 56.8% (51.7% to 61.7%) |

PTED prevalence was calculated by percentage scoring above a mean total score of 1.6, 2 and 2.5.
Each MIES item was coded as endorsed if the participant responded either 'slightly agree', 'moderately agree' or 'strongly agree'. Subscale prevalence was calculated by endorsement of any item within that subscale.
MIES, Moral Injury Events Scale; PTED, post-traumatic embitterment disorder.

incremental fit indices close to 0.90 and absolute fit indices close to 0 (ie, <0.06) represented adequate model fit. Analyses were carried out using IBM SPSS V.26, AMOS V.26 or R V.4.1.0.

## RESULTS
Of the 400 responses recorded, there was <1% missing data for social desirability and professional role, and 23.8% missing data for ethnicity, which was missing completely at random, $\chi(15)=7.737$, p=0.934.[30]

### Prevalence of MIES, PTED and exposure to occupational stressors
In this sample, 72.3% (95% CI 67.6% to 76.6%) experienced at least one MIE. On the PTED Scale, using the clinical cut-off score of $\geq 2$[11] 19.3% (95% CI 15.5% to 23.5%) of the sample were embittered (see table 1). For item level prevalence estimates, see online supplemental table 2. The average score on the MIES was 2.55 (SD=1.13) with the highest scores observed on the transgressions-others subscale, followed by betrayal and transgressions-self (see

**Table 2** Descriptive statistics, normality estimates and Cronbach's alpha estimates

| | Mean±SD | Skew | Kurtosis | Cronbach's |
|---|---|---|---|---|
| Age | 36.84±10.73 | 0.73 | −0.22 | |
| Years* | 4.50 (2, 9) | 2.10 | 5.14 | |
| Social desirability | 6.93±2.80 | −0.07 | −0.50 | 0.71 |
| PTED | 1.12±0.90 | 0.45 | −0.75 | 0.96 |
| Transgressions-others | 2.98±1.48 | 0.16 | −1.16 | 0.78 |
| Transgressions-self | 2.17±1.22 | 0.80 | −0.37 | 0.93 |
| Betrayal | 2.77±1.38 | 0.35 | −0.93 | 0.80 |
| Optimism | 6.60±2.72 | −0.46 | −0.39 | 0.83 |
| Pessimism | 5.82±2.82 | 0.16 | −0.61 | 0.84 |
| Resilient coping style | 14.60±2.40 | −0.50 | 0.94 | 0.62 |
| Distributive justice | 18.75±4.70 | −0.63 | −0.13 | 0.89 |
| Procedural justice | 20.33±4.77 | −0.98 | 0.84 | 0.95 |
| Occupational stressors | 9.91±3.44 | 0.71 | −0.06 | 0.76 |
| CFC-I | 23.47±7.68 | 0.19 | −0.61 | 0.85 |
| CFC-F | 34.58±6.85 | −0.50 | 0.41 | 0.84 |
| Self-esteem | 28.18±5.80 | −0.25 | −0.21 | 0.91 |

*Median/IQR (Q1, Q3) reported due to non-normality.
betrayal, moral injury betrayal; CFC-F, consideration of future consequences-future; CFC-I, consideration of future consequences-immediate; PTED, post-traumatic embitterment disorder; transgressions-others, moral injury transgression by others; transgressions-self, moral injury transgression by self.

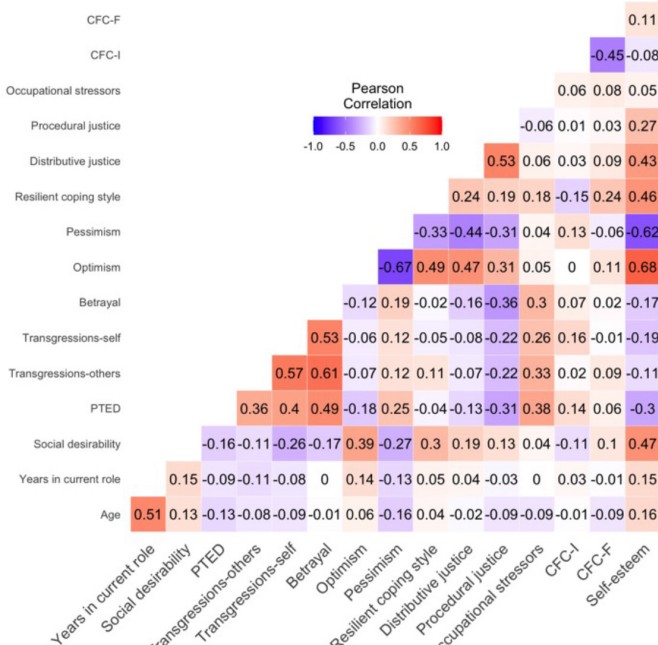

**Figure 1** Heatmap showing Pearson's correlation coefficients of study variables. CFC-F, consideration of future consequences-future; CFC-I, consideration of immediate consequences-immediate; PTED, post-traumatic embitterment disorder.

table 2). Prevalence of exposure to occupational stressors at 'occasionally' or above on the exposure scale revealed that 69.8% of participants reported being exposed to COVID-19 in their job, 57.5% were affected by a lack of PPE or COVID-19 training, 56.8% reported making difficult resource allocation decisions, 56.3% reported feeling unable to provide adequate care or save patients they usually would and 64% reported experiencing separation from, or fear for, loved ones due to working during the pandemic.

### Univariate association between MIES scores, PTED scores and predictor variables
Table 2 displays mean scores and internal consistency estimates, and figure 1 displays a Pearson's correlation heatmap of continuous study variables. Online supplemental table 1 displays univariate tests of differences between categorical demographic variables and PTED and MIES scores.

PTED was positively associated with all subscales of the MIES, and the largest association was with betrayal (p's<0.001, see figure 1). These associations approached moderate effect sizes (r=0.36–0.49) indicating that PTED and MIEs are distinct, but related psychological constructs.

PTED was associated with older age (p=0.011) and current self-reported mental health diagnoses (p<0.001), transgressions-others was associated with fewer years in current role (p=0.033), betrayal was associated with current self-reported mental health diagnoses (p=0.013) and transgressions-self was associated with professional

role (p=0.021) though no roles were significantly different from each other when applying Games-Howell correction for multiple comparisons. Risk factors for all dependent variables included exposure to occupational stressors (p's<0.001) and pessimism (p<0.001 with PTED and betrayal, p=0.014 with transgressions-others and p=0.015 with transgressions-self). CFC-I was a risk factor for PTED (p=0.005) and transgressions-self (p=0.001), and resilient coping style was a risk factor for transgressions-others (p=0.035). Protective factors for all dependent variables included self-esteem (p<0.001 for associations with PTED, transgressions-self, and betrayal, and p=0.027 with transgressions-others) and procedural justice (p's<0.001). Distributive justice and optimism were protective factors of PTED (distributive justice p=0.009, optimism p<0.001) and of betrayal (distributive justice p=0.002, optimism p=0.015). All dependent variables were influenced by social desirability (p's from <0.001 to 0.032, see figure 1 and online supplemental table 1). All effect sizes were below the recommended minimum practical effect size (RMPE) or were small sized.

### Hierarchical multiple regressions
All variables from Univariate (unadjusted) analyses were included in hierarchical regressions, except for gender and ethnicity, which were not significantly associated with any outcome variables (see table 3 for regression model summary and online supplemental tables 3 and 4 for regression coefficients).

The full model including demographics, occupational stressors and personality characteristics explained between 38% and 22% of the variance in PTED and MIES subscales scores (adjusted $R^2$=35%–18%). This represents a large effect for PTED ($f^2 \geq 0.35$) and moderate effect sizes for all MIES regressions ($f^2 \geq 0.15$). Demographics significantly explained 5%-10% of the variance in PTED, transgressions-self and transgressions-others scores (p<0.05), exposure to occupational stressors in step two significantly explained a further 8%–16% of variances in all four regressions (p<0.001), and personality characteristics in step three significantly explained a further 6%–12% of variance in all four regressions (p<0.001), with small-to-moderate effect sizes ($f^2$'s of 0.05–0.21).

Statistically significant risk factors for PTED included being in a clinical support role compared with an allied health (p=0.036) or primary care practitioner (p=0.030), having current mental health diagnoses (p=0.006), exposure to occupational stressors (p=0.006), CFC-I (p=0.012), CFC-F (p=0.036) and distributive justice (p=0.024). Statistically significant protective factors included belief in procedural justice and self-esteem (p's=0.006).

Statistically significant risk factors for transgressions-others included being in a clinical support role compared with allied health professionals (p=0.020), exposure to occupational stressors (p=0.002) and resilient coping style (p=0.014). The only statistically significant protective factor was belief in procedural justice (p=0.002).

**Table 3** Model summary of hierarchical regression analyses using all possible predictors and only significant predictors of PTED, transgressions-others, transgressions-self and betrayal scores with 95% bias corrected and accelerated CIs (2000 samples) (N=394–396)

| | PTED† | Transgressions-others† | Transgressions-self† | Betrayal† |
|---|---|---|---|---|
| Model including all possible predictors | | | | |
| Step one—controls | 0.10*** | 0.04 | 0.10*** | 0.05* |
| Step two—occupational stressors | 0.26/0.16*** | 0.16/0.12*** | 0.17/0.08*** | 0.15/0.10*** |
| Step three—personality | 0.38/0.12*** | 0.22/0.06*** | 0.24/0.07*** | 0.26/0.10*** |
| Model including only significant predictors from the previous model | | | | |
| | PTED† | Transgressions-others† | Transgressions-self‡ | Betrayal‡ |
| | $R^2/\Delta R$ | $R^2/\Delta R$ | $R^2/\Delta R$ | $R^2/\Delta R$ |
| Step one—controls | 0.08*** | 0.02 | 0.10*** | 0.04* |
| Step two—occupational stressors | 0.24/0.16*** | 0.13/0.14*** | 0.17/0.08*** | 0.14/0.10*** |
| Step three—personality | 0.37/0.13*** | 0.19/0.04*** | 0.21/0.04*** | 0.24/0.10*** |

**p<0.05, **p<0.01, ***p<0.001.
†N=396.
‡N=394.
betrayal, Moral injury betrayal; PTED, post-traumatic embitterment disorder; transgressions-others, Moral injury transgression by others; transgressions-self, Moral injury transgression by self.

Statistically significant risk factors for transgressions-self included being in a clinical support role compared with an allied health (p=0.002) and managerial role (p=0.028), exposure to occupational stressors (p=0.002) and CFC-I (p=0.037). Statistically significant protective factors included belief in procedural justice (p=0.002) and self-esteem (p=0.016).

Statistically significant risk factors for betrayal were being in a clinical support role compared with an allied health role (p=0.036) and exposure to occupational stressors (p=0.006), whereas belief in procedural justice was a statistically significant protective factor (p=0.006). Social desirability was significantly associated with lower reporting of betrayal (p=0.012) and transgressions-self (p=0.002).

Overall, predictors with βs of practical significance (≥0.20) included exposure to occupational stressors as a risk factor and belief in procedural justice as a protective factor for PTED and all MIES subscales, self-esteem as a protective factor for PTED, and social desirability resulted in lower transgressions-self scores (see online supplemental table 3). When running each hierarchical regression with only significant variables to improve model precision and parsimony, each full model explained approximately the same amount of variance in PTED and MIES scores ($R^2$=19%–37%, see table 3). Managerial role (p=0.064) and self-esteem (p=0.427) become non-significant in the transgressions-self model (see online supplemental table 4).

## DISCUSSION

This is the first study to show that both PTED and MIEs were prevalent in UK HSCWs during September–October 2020 of the COVID-19 pandemic. Correlations show that PTED and MIEs are distinct but related constructs and are significantly associated with demographic, occupational and personality variables. Hierarchical regressions show that occupational stressors were a key risk factor, and personal belief in a procedurally just world, a key protective factor for both constructs. To a lesser extent, being in a clinical support role was a risk factor for both PTED, and exposure to MIEs. Beyond this, risk and protective factors had more selective effects. For example, in the final regression models, risk factors for PTED included having current self-reported mental health diagnoses, higher consideration of the immediate and future consequences of actions, and higher levels of personal belief in distributive justice. Self-esteem was a resiliency factor. Higher consideration of the immediate consequences of actions was a risk factor for transgressions-self, and higher levels of resilient coping style was a risk factor for transgressions-others. Social desirability was associated with significantly lower reporting of transgressions-self and betrayal, though effect sizes indicate these selective effects were less practically meaningful.

Our study has several strengths and limitations. This study is novel, providing the first preliminary evidence of both MIEs and PTED in UK HSCWs during the COVID-19 pandemic. The results are also generalisable. The survey extends research on HCWs by being open to all UK HSCWs on Prolific, and sample characteristics were broadly similar to the wider NHS workforce in terms of gender (female 76%, NHS=77%) and ethnicity given that we had missing data (white=63%, ethnic minorities=13.3%, NHS white=79.2% vs ethnic minorities=20.7%). The sample included diverse job roles, and 25% of the sample self-reported current mental health diagnoses, which matches the general population. The

most notable limitation is that cross-sectional baseline data were analysed, and only future analyses from our longitudinal study can establish causal relationships. Data were also collected several months after the pandemic began, no comparisons to prepandemic levels of PTED and MIEs were possible, and despite controlling for social desirability, surveys suffer from response and social desirability bias.

Despite these limitations, our study supports early findings that healthcare staff believe themselves to have been exposed to MIEs during the pandemic.[14] Prevalence in this sample is higher than that reported in US HCWs during COVID-19,[12] and is similar to those reported by active-duty US marines.[18] Prevalence of PTED in this sample is eight times higher than a general population sample,[11] but is lower than reported by UK NHS staff attending an occupational health centre pre pandemic.[31] This indicates that healthcare staff experiencing PTED may be at a higher risk of requiring occupational health services. Further, despite similarities in aetiology and symptomology, studies have rarely examined the overlap between PTED and moral injury. Moderate correlations provide early support to our argument that MIEs can trigger PTED in some individuals and future longitudinal analyses will investigate this model.

Importantly, MIEs and PTED shared the same key risk and protective factors. Experiencing MIEs and PTED was more likely in clinical support workers and those experiencing occupational stressors. Exposure to occupational stressors likely increases the frequency of MIEs that can cause embitterment. Personal belief in procedural justice, which is the belief that you experience fair processes protected against exposure to MIEs and PTED. While this belief is generally seen as a disposition that can protect against the negative impact of injustices, it is equally possible that belief in a just world can be modified by exposure to injustices, leading to PTED.

Other vulnerability and resiliency factors vary. Having current mental health diagnoses increased the risk of PTED, and mental health conditions are often comorbid. Considering immediate and future consequences of actions increased risk of PTED, and this may be explained by feelings of helplessness when thinking about the unjust life event, as noted in diagnostic criterion.[10 11] Belief in distributive justice, which is the belief that you experience fair outcomes increased risk of PTED in contrast to findings that procedural justice, was a resiliency factor. Surprisingly, resilient coping style increased exposure to transgressions by others, despite findings that resilient coping tendencies are linked with better psychological well-being.[23] It is possible that people feel more comfortable committing moral transgressions around individuals that they consider to be resilient. Self-esteem increased resilience to PTED, and reliably buffers against poor mental health in the literature.[22] Combined, these findings indicate several vulnerability and resiliency factors that either increase or decrease the risk of exposure to MIEs, and the mental health impact in the form of PTED.

Having said that, regressions show these variables only partially predict PTED and morally injurious experiences (19%–37%) and so support for theoretical models highlighting their importance is limited.[8 15 16]

These findings allow for several recommendations. First, some individuals may recover from moral injuries and feelings of embitterment on their own, but others will need psychological support and clinicians should be aware that these individuals are relatively treatment resistant and have different treatment needs.[8 10 11] Self-care strategies should focus on increasing self-esteem and reducing tendencies to think about the consequences of one's actions, in line with this study's findings. More importantly though, there should be an emphasis not on the individual to 'recover' but on healthcare leaders to tackle systemic issues that are causing moral injuries and embitterment in the first place. In this study, staff across all levels of health and social care have reported feeling betrayed, morally violated and that they had been treated unjustly and unfairly during the pandemic. A proportion of these responses have been directly linked to the working environment (eg, lack of PPE and resources). Healthcare authorities should strive to maintain fair 'processes' even if reducing all workplace stressors is unachievable. At a time when the UK and other countries need a healthy workforce to respond to the pandemic and its aftermath, staff retention is a priority.

To strengthen prevalence estimates reported here, large-scale cohort studies using randomly selected, representative samples are needed. Follow-up data collection with these samples can establish whether prevalence estimates increase across time as the pandemic continues. Future studies should identify the cognitive, affective and behavioural manifestations of PTED and moral injury in healthcare professionals to inform treatment and treatment efficacy then needs to be established. Although beyond the scope of this paper, future studies should investigate the overlap between PTED and moral injury symptoms using newly developed symptom scales, and their overlap with other similar constructs like burnout.[9]

In conclusion, we are among the first to provide prevalence estimates of exposure to MIEs and PTED in HSCWs as a direct result of working during the COVID-19 pandemic. We provide evidence that those who are clinical support staff, exposed to occupational stressors and have a lack of belief that they experienced fair processes are among the most at risk. These findings indicate that measures to reduce workplace stressors and ensuring staff feel they receive fair procedures at work should be a priority of healthcare organisations.

**Contributors** CJB designed the project and data collection tools, collected the data, cleaned, analysed and interpreted the data and drafted and revised the paper. She is guarantor. MTM analysed and interpreted the data and revised the paper. JCC designed the project and data collection tools, interpreted the data and revised the paper. All authors gave final approval for the publication of this manuscript. The corresponding author attests that all listed authors meet authorship criteria and that no others meeting the criteria have been omitted.

**Funding** CJB was supported by the Economic and Social Research Council UK (ES/P000665/1) and this work received COVID-19 strategic funding from the University of Liverpool.

**Competing interests** The lead author had funding support from the Economic and Social Research Council and University of Liverpool COVID-19 strategic funding for the submitted work

**Patient and public involvement** Patients and/or the public were not involved in the design, or conduct, or reporting, or dissemination plans of this research.

**Patient consent for publication** Not applicable.

**Ethics approval** This study involves human participants and was approved by Ethics Committee at The University of Liverpool (7862). Participants gave informed consent to participate in the study before taking part.

**Provenance and peer review** Not commissioned; externally peer reviewed.

**Data availability statement** Data are available upon reasonable request. Data from this study will be made available upon reasonable request by contacting the corresponding author at hlcbrenn@liverpool.ac.uk. Data will be made available after an embargoed period of 12 months to permit planned analyses of the longitudinal dataset and will be available for 10 years in line with ethical considerations.

**ORCID iD**
Chloe J Brennan http://orcid.org/0000-0002-8284-835X

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
