## [Reviewer comments · BMJ Open]

ARTICLE DETAILS

TITLE (PROVISIONAL)	Morally injurious events and post-traumatic embitterment disorder in UK health and social care professionals during COVID-19: a cross sectional web survey
AUTHORS	Brennan, Chloe; McKay, Michael; Cole, Jon

VERSION 1 – REVIEW

REVIEWER	Faisal Akram Saint Elizabeths Hospital/ DC Department of Behavioral Health, Psychiatry
REVIEW RETURNED	22-Jun-2021

GENERAL COMMENTS	- This study is timely and well-thought of. Text is clear, coherent, and organized. Study design is sound. The manuscript will benefit greatly from: (1) Explanation of why moral injury and PTED were explored together: It appears that the authors used the term moral injury exclusively to specify whether an event was perceived as morally-injurious or not. However, common usage of moral injury is inclusive of both morally-injurious event and its mental health impact. There is now a moral-injury symptoms scale (see PMID: 32681398) and the term itself is being regarded as an "unofficial" psychiatric disorder by many. Latter conceptualization is not captured by the Moral Injury Events Scale (MIES), the first instrument developed to measure exposure to military events that could produce moral injury symptomatology (Nash et al., 2013). If moral injury and PTED were considered distinct, then one limitation of study is that it did not measure moral injury symptomatology. One solution would be to replace "moral injury" with "moral injury events" to accurately highlight that MIES only measured moral injury events. In addition, PTED could be viewed as embitterment after moral injury event in my opinion (with the limitation that temporality was not measured). Viewing these as separate decrease the conceptual validity of study. Regardless, readers would benefit from knowing the authors' rationale behind simultaneous exploration of both moral injury and PTED. (2) Addressing the issue of symptom overlap with burnout in moral injury research: Excessive workload leads to burnout with similar symptoms except that no moral injury event took place. There is an ongoing debate whether some burnout is disguised moral injury (PMID: 31571807). Since COVID-19 caused increased workload, the study would benefit from highlighting the overlap between burnout and moral injury, and whether it was controlled for or not in the analysis.
---

	(3) Discussion of significant results: The results calls for a discussion of why some variables such as being in a clinical support role compared to an allied health or primary care practitioner, having an existing mental health diagnoses, exposure to occupational stressors, CFC-I, CFC-F, and personal BJW-DJ increase the risk of PTED. In my opinion, these variables point to vulnerability factors, i.e. making individuals at risk of either increased frequency of moral injury events or their mental health impact. Similarly, there is an emphasis on "perception" of transgressions in the interpretation of results. For example, occupational stressors increase the risk of betrayal. What could be the mechanism? Do occupational stressors increase the frequency of events that could lead to betrayal, or is it that occupational stressors make individuals more sensitive to betrayal? (4) Page 25, line 10: "Moral injury is a likely consequence of acute exposure to the pandemic." Please explain this statement. (5) Page 26, line 3-6: "Managers must use fair processes towards their staff, otherwise they risk a demoralised workforce, high turnover rates, and a culture that no one wants to work in." This statement puts a lot of moral burden on managers. On the contrary, healthcare across the world grappled with an accurate delineation of what was fair and what was not during the COVID pandemic (PMID: 33778145).
--	---

REVIEWER	Danielle Glick University of Maryland School of Medicine, Pulmonary, Critical Care, and Sleep Medicine
REVIEW RETURNED	13-Aug-2021

GENERAL COMMENTS	This manuscript addresses an important and currently under-recognized challenge in the healthcare community, namely the effects of moral injury and post-traumatic embitterment. While this is likely not a new entity, the specific pressures, both societal and professional, imposed by the COVID-19 pandemic have raised global concerns over the effects of these injuries on healthcare workers. Overall, the authors do a nice job of identifying the presence of PTED and moral injury in a population of UK health and social care workers. They go further by identifying risk factors for each of these injuries. However, the main message (that moral injury and PTED are prevalent and of concern) is hindered by the breadth of analysis presented in the manuscript. Would recommend restructuring the manuscript to improve clarity of the message, with modifications as follows:  - The main hypothesis/aim of the study is "to establish the prevalence of PTED and moral injury and identify potential risk and protective factors". To better support the first part of this statement, would recommend at a minimum to reverse the order of tables 1 and 2, and to consider instead generating a novel table to only address the prevalence of both PTED and moral injury; this is somewhat accomplished in table 2, however it is very difficult to interpret the prevalence from this table alone. Would instead:  - present the overall scores on the scales (19% PTED and 73% moral injury) in the context of a general demographics table (restructure table 2 to either be more general or to include a "group overall" column) - consider presenting the data in table 1, and perhaps, table 2 as supplementary
--

	- can you clarify the roles further? health vs social care worker? With the above modifications, the message might be clearer and more succinct. - please define thresholds of abnormality on the three subscales in the MIES (Materials, paragraph 2) - Consider relocating the "preliminary results" discussion paragraph to the data analysis portion of the methods section. While important statistically, the presence of this information in the results section detracts from the impact of the next section "prevalence of MIES, PTED, and exposure to occupational stressors" which is the main objective of the study. - The hierarchical multiple regressions offer important insight into the risk and protective factors for development of moral injury and PTED. However, the volume of statistical analysis and scales used makes this somewhat confusing. A reader without familiarity of the scales used may find interpretation of the results difficult. Could the results (tables 3 and 4) be either condensed to the most important outcomes and/or placed in supplemental data with presentation in the results section with fewer acronyms? - Similarly, in paragraph one of the discussion, would recommend removing or defining acronyms in lay terms for ease of interpretation
--	--

REVIEWER	Snehil Gupta AIIMS Bhopal, Department of Psychiatry
REVIEW RETURNED	05-Oct-2021

GENERAL COMMENTS	Dear authors, I congratulate you all for conducting a good research work concerning the COVID-19 pandemic and psychological health of the HCWs. Here are my few comments on the manuscript: Title: Does not talk about the study design. It should be the part of the title to make it more comprehensive. Abstract: whereas personal belief in a procedurally just world was a protective mechanism (slight elaboration would be useful for the readers). Strengths and limitations: Sentences need to be brief (one liner) as per the journals' instruction. Main file Introduction- The sentence-"adding to work-related stress that prior to the pandemic was at its highest level since 2014.[2]" needs simplification. Methodology- Better term would be 'mean total score' (of >2) than mean score as provided in the original scale by Linden, 2013 Statistics: Though the statistical analyses seem fine to me, however, I would appreciate if a biostatistician can comment upon the over analytical design of the study for reviewing it more robustly. Ethical issues: needs greater consideration such as what if a particular participant experienced considerable moral injury or PTED, were there any provision of mental health support for them?. How their psychological distress addressed or intended to address? Results: Well-written Discussion: The relationship between the moral injury and PTED
---

	and dependent variables with the scores on MI & PTED should be elaborated in line of psychological construct and available literature. 23-24 despite stark similarities, I think author means to say stark “dissimilarities” Conclusion: fine Reference: many of the references missing doi. Please add them wherever indicated. Final comments: Overall, a well conducted study and written manuscript. However, minor changes can be made to improve the quality of the paper.
--	---

REVIEWER	Fahmida Hossain Duquesne University
REVIEW RETURNED	05-Feb-2022

GENERAL COMMENTS	Manuscript: bmjopen-2021-054062 Moral injury and Post-traumatic embitterment disorder in UK health and social care professionals during COVID-19: a cross sectional analysis Reviewer(s)' Comments to Author: The purpose of this paper is to find moral injury and post-traumatic embitterment disorder (PTED) in UK health and social care professionals during the COVID-19 pandemic. This is one of the first empirical research conducted on this topic. I believe this is a fantastic work, and this will help enrich the Covid literature. However, I have some specific comments and recommendations.  - The author assumed everyone who will read this has a great understanding of Moral Injury, which is not guaranteed. In fact, Moral injury is used for the military, which created some arguments. In addition, some scholar still believes the term Moral Injury is not suitable for the medical field. So, I recommend adding a bit more on Moral Injury. Give a bit more detail about Moral Injury, and how these fits into the Covid crisis. - I also recommend adding the concept of embitterment. What is that? How does it relate, and how is it different? It will be helpful to have that at the beginning of the paper before methods. - Make sure to use the appropriate term. In conclusion, you mentioned “manager”. Who are they? Do you mean the healthcare authorities/ administration? Then explain that. A manager is not the right word! - This paper lacks a reasonable conclusion. Add more details to the conclusion to let it stand out. - I would highly recommend is to address the importance of Self-care along with organizational support. However, relying on administrative support only will bring more damage to mental health. Instead, some proactive self-care strategies should be implemented. - Overall, I think this is a strong paper. Again, it is the starting of many more empirical research on Covid-19.
---

REVIEWER	Charles W. Goss Washington University in St Louis, Biostatistics
REVIEW RETURNED	10-Feb-2022

GENERAL COMMENTS

Thank you for the opportunity to review this paper that reports results from a survey of health and social care workers during the COVID-19 pandemic in the fall of 2020. This paper addresses moral and post-traumatic embitterment disorder (PTED) in response to the pandemic that is not widely reported in the literature. Overall, this is a well-written manuscript and a well-done study. I have some comments and suggestions aimed at improving the clarity and statistical reporting of the manuscript.

1. This study took place in the fall of 2020 and there was a proposed follow-up 1 year later. I can appreciate that a baseline paper such as this can be critical to get published prior to completion of the study, but it has been > 1 year since these data were collected and the manuscript indicates that longitudinal data should have been collected in the fall of 2021. Please provide an explanation for why longitudinal data were not included in this manuscript.
2. The results subsection entitled "Preliminary results" is somewhat misleading as this oftentimes refers to non-definitive results from small studies (e.g., early-phase clinical trials). I think it could be removed or renamed more appropriately. Additionally, this section has very detailed descriptions of model diagnostics. While I appreciate the level of detail included, I think text could be made much more concise and some of the material could even be included in the methods.
3. Table 1: I would make separate tables for the descriptive statistics (mean, SD) and the Pearson correlation coefficient estimates.
4. Table 1: This table contains a lot of variables and is a bit challenging to interpret. I would recommend creating a correlation matrix heatmap. See this website for examples with various software packages: Heatmap Colored Correlation Matrix | LOST (lost-stats.github.io)
5. Page 12, lines 3-16: "Significantly" could be dropped altogether in the first sentence. As an example, "PTED was associated with older age ($P = XXX$) and existing self-reported mental health diagnoses ($P = XXX$)..."
6. There are several citations that are included in both the results and the methods (e.g., Cohen, Ferguson, Little). The citations are only needed in the methods.
7. Ferguson is misspelled in the methods and results as "Fergusson". The year that this article was published is 2009 not 2016 as listed in the references.
8. Specific significant (or not significant) variables in the results text should include associated P values.
9. Rounding is inconsistent for percentages in the results. Sometimes there are 2 decimal places and sometimes 1. I would use a single decimal place.
10. Results should be formatted with a "0" before the decimal point if the value is < 1 (including P values).
11. Page 6, line 37: Define "small to moderate effect size".
12. Page 7, lines 20-21: The way this is phrased is confusing because the previous sentence references a factor analysis, but I believe that the "factor" in this sentence is not a factor score. Please clarify.
13. Page 11: remove "(two tailed)"
14. Page 23, lines 49-54: An important limitation is that only baseline (cross sectional) data were analyzed in this paper. This is clearly stated in the abstract, but it is not clear in the discussion.
15. The acronym NHS is not defined.

VERSION 1 – AUTHOR RESPONSE

Reviewer: 1

Dr. Faisal Akram, Saint Elizabeths Hospital/ DC Department of Behavioral Health

Comments to the Author:

- This study is timely and well-thought of. Text is clear, coherent, and organized. Study design is sound. The manuscript will benefit greatly from:

(1) Explanation of why moral injury and PTED were explored together: It appears that the authors used the term moral injury exclusively to specify whether an event was perceived as morally-injurious or not. However, common usage of moral injury is inclusive of both morally-injurious event and its mental health impact. There is now a moral-injury symptoms scale (see PMID: 32681398) and the term itself is being regarded as an "unofficial" psychiatric disorder by many. Latter conceptualization is not captured by the Moral Injury Events Scale (MIES), the first instrument developed to measure exposure to military events that could produce moral injury symptomatology (Nash et al., 2013). If moral injury and PTED were considered distinct, then one limitation of study is that it did not measure moral injury symptomatology. One solution would be to replace "moral injury" with "moral injury events" to accurately highlight that MIES only measured moral injury events. In addition, PTED could be viewed as embitterment after moral injury event in my opinion (with the limitation that temporality was not measured). Viewing these as separate decrease the conceptual validity of study. Regardless, readers would benefit from knowing the authors' rationale behind simultaneous exploration of both moral injury and PTED.

Unfortunately, the moral injury symptom scale (PMID: 32681398) had not been developed prior to the study commencing so the MIES was considered the best scale as symptom scales were military specific.

We have addressed this in multiple ways:

- 1. In the introduction we have explained the difference between morally injurious events (MIEs) and symptoms.**
- 2. In the materials section we have added to the MIES scale to explain that a higher score indicated more exposure to morally injurious events.**
- 3. We have replaced the phrase 'moral injury' with morally injurious events (MIEs) in the title and throughout.**
- 4. We have added to the introduction to explain the rationale behind exploring both moral injury and PTED and argue that MIEs could lead to PTED. This will be explored with longitudinal analyses which is noted at the end of the introduction and in the discussion.**
- 5. In the discussion section we have noted that future studies could look at the overlap between moral injury symptomatology and PTED.**

(2) Addressing the issue of symptom overlap with burnout in moral injury research: Excessive workload leads to burnout with similar symptoms except that no moral injury event took place. There is an ongoing debate whether some burnout is disguised moral injury (PMID: 31571807). Since COVID-19 caused increased workload, the study would benefit from highlighting the overlap between burnout and moral injury, and whether it was controlled for or not in the analysis.

In paragraph three of the introduction, we have noted that some researchers argue burnout is disguised moral injury, and in the future studies section of the discussion we suggested that future studies should investigate their relationship. We mentioned that this was beyond the scope of this paper.

(3) Discussion of significant results: The results calls for a discussion of why some variables such as being in a clinical support role compared to an allied health or primary care practitioner, having an existing mental health diagnoses, exposure to occupational stressors, CFC-I, CFC-F, and personal BJW-DJ increase the risk of PTED. In my opinion, these variables point to vulnerability factors, i.e. making individuals at risk of either increased frequency of moral injury events or their mental health impact.

In paragraphs four and five of the discussion we have expanded on why some variables increased/decreased the risk of PTED and/or MIEs and noted that this points to vulnerability and resiliency factors.

Similarly, there is an emphasis on "perception" of transgressions in the interpretation of results. For example, occupational stressors increase the risk of betrayal. What could be the mechanism? Do occupational stressors increase the frequency of events that could lead to betrayal, or is it that occupational stressors make individuals more sensitive to betrayal?

We have added to the discussion to explain that exposure to occupational stressors likely increases frequency of MIEs that can cause embitterment.

(4) Page 25, line 10: "Moral injury is a likely consequence of acute exposure to the pandemic." Please explain this statement.

We have changed the sentence to "Despite these limitations, our study supports early findings that healthcare staff feel they have been exposed to MIEs during the pandemic".

(5) Page 26, line 3-6: "Managers must use fair processes towards their staff, otherwise they risk a demoralised workforce, high turnover rates, and a culture that no one wants to work in." This statement puts a lot of moral burden on managers. On the contrary, healthcare across the world grappled with an accurate delineation of what was fair and what was not during the COVID pandemic (PMID: 33778145).

We agree that managers are also experiencing difficult moral decisions. In the discussion 'managers' has been replaced with 'healthcare authorities' to remove blame from individuals. We have also removed the sentence about turnover and culture.

Reviewer: 2

Dr. Danielle Glick, University of Maryland School of Medicine

Comments to the Author:

This manuscript addresses an important and currently under-recognized challenge in the healthcare community, namely the effects of moral injury and post-traumatic embitterment. While this is likely not

a new entity, the specific pressures, both societal and professional, imposed by the COVID-19 pandemic have raised global concerns over the effects of these injuries on healthcare workers. Overall, the authors do a nice job of identifying the presence of PTED and moral injury in a population of UK health and social care workers. They go further by identifying risk factors for each of these injuries. However, the main message (that moral injury and PTED are prevalent and of concern) is hindered by the breadth of analysis presented in the manuscript. Would recommend restructuring the manuscript to improve clarity of the message, with modifications as follows:

- The main hypothesis/aim of the study is "to establish the prevalence of PTED and moral injury and identify potential risk and protective factors". To better support the first part of this statement, would recommend at a minimum to reverse the order of tables 1 and 2, and to consider instead generating a novel table to only address the prevalence of both PTED and moral injury; this is somewhat accomplished in table 2, however it is very difficult to interpret the prevalence from this table alone. Would instead:

- present the overall scores on the scales (19% PTED and 73% moral injury) in the context of a general demographics table (restructure table 2 to either be more general or to include a "group overall" column)

Table 1 in the main manuscript now includes the prevalence estimates, and supplementary table 2 includes further detailed prevalence estimates per item.

- consider presenting the data in table 1, and perhaps, table 2 as supplementary

Table 2 from the original paper is now table 1 in the supplementary materials.

- can you clarify the roles further? health vs social care worker?

The sample was coded into categories in line with the NHS website careers website. We have provided a reference to the NHS website in the participants and procedures section now.

With the above modifications, the message might be clearer and more succinct.

- please define thresholds of abnormality on the three subscales in the MIES (Materials, paragraph 2)

No thresholds exist in the literature, but we have defined how we calculated prevalence estimates in the materials section (MIES paragraph 2).

- Consider relocating the "preliminary results" discussion paragraph to the data analysis portion of the methods section. While important statistically, the presence of this information in the results section detracts from the impact of the next section "prevalence of MIES, PTED, and exposure to occupational stressors" which is the main objective of the study.

We have integrated the preliminary results paragraph into the data analysis methods section.

- The hierarchical multiple regressions offer important insight into the risk and protective factors for development of moral injury and PTED. However, the volume of statistical analysis and scales used makes this somewhat confusing. A reader without familiarity of the scales used may find interpretation of the results difficult. Could the results (tables 3 and 4) be either condensed to the most important outcomes and/or placed in supplemental data with presentation in the results section with fewer acronyms?

In the text, tables 3 and 4 have been replaced with one table (table 3) which is the model summary of the overall variance explained by the regressions. The coefficients can be found in supplementary tables 3 and 4. We have removed some of the acronyms from the paper altogether (i.e., R-SES to self-esteem, BRCS to resilient coping style, personal BJW-DJ to distributive justice and personal BJW-PJ to procedural justice).

- Similarly, in paragraph one of the discussion, would recommend removing or defining acronyms in lay terms for ease of interpretation

In the discussion section we have avoided using acronyms where possible.

Reviewer: 3

Dr. Snehil Gupta, AIIMS Bhopal

Comments to the Author:

Dear authors, I congratulate you all for conducting a good research work concerning the COVID-19 pandemic and psychological health of the HCWs. Here are my few comments on the manuscript:

Title: Does not talk about the study design. It should be the part of the title to make it more comprehensive.

We have changed the study title to include the design i.e., cross-sectional web survey.

Abstract: whereas personal belief in a procedurally just world was a protective mechanism (slight elaboration would be useful for the readers).

We have added to the abstract to explain that belief in a procedurally just world is the belief that you personally experience fair processes.

Strengths and limitations:

Sentences need to be brief (one liner) as per the journals' instruction.

We have shortened the strengths and limitations to shorter, one-line sentences.

Main file

Introduction-

The sentence-"adding to work-related stress that prior to the pandemic was at its highest level since 2014.[2]" needs simplification.

This sentence has been re-worded to "In the year prior to the pandemic, U.K healthcare workers (HCWs) reported work-related stress at its highest level since 2014."

Methodology-

Better term would be 'mean total score' (of >2) than mean score as provided in the original scale by Linden, 2013

We have re-worded this to mean total score of >2 in line with your suggestion.

Statistics: Though the statistical analyses seem fine to me, however, I would appreciate if a biostatistician can comment upon the over analytical design of the study for reviewing it more robustly.

We have made revisions in the results section in line with all reviewers' suggestions.

Ethical issues: needs greater consideration such as what if a particular participant experienced considerable moral injury or PTED, were there any provision of mental health support for them?. How their psychological distress addressed or intended to address?

In the participants and procedure section we have explained how we supported participants that may have experienced considerable levels of moral injury and/or PTED. This was by signposting to appropriate support, and to researchers contact details on the information and debrief sheets.

Results: Well-written

Discussion: The relationship between the moral injury and PTED and dependent variables with the scores on MI & PTED should be elaborated in line of psychological construct and available literature. 23-24 despite stark similarities, I think author means to say stark "dissimilarities"

We did mean to say 'similarities' and have explained in more detail what these similarities are in the introduction now so hopefully it is clearer why. In the discussion the relationship between PTED and MI, and their relationship to the other scales has been elaborated on in line with existing literature.

Conclusion: fine

Reference: many of the references missing doi. Please add them wherever indicated.

We have now included DOI wherever possible.

Final comments:

Overall, a well conducted study and written manuscript. However, minor changes can be made to improve the quality of the paper.

Reviewer: 4

Dr. Fahmida Hossain, Duquesne University

Comments to the Author:

Overall, I think this is a strong paper. It is the starting of many more empirical research on Covid-19. But this needs some minor revision before approval. [Please see attached document]

The purpose of this paper is to find moral injury and post-traumatic embitterment disorder (PTED) in UK health and social care professionals during the COVID-19 pandemic. This is one of the first empirical research conducted on this topic. I believe this is a fantastic work, and this will help enrich the Covid literature.

However, I have some specific comments and recommendations.

- The author assumed everyone who will read this has a great understanding of Moral Injury, which is not guaranteed. In fact, Moral injury is used for the military, which created some arguments. In addition, some scholar still believes the term Moral Injury is not suitable for the medical field. So, I recommend adding a bit more on Moral Injury. Give a bit more detail about Moral Injury, and how these fits into the Covid crisis.

In paragraphs three, four and five of the introduction we have provided more detail on moral injury.

- I also recommend adding the concept of embitterment. What is that? How does it relate, and how is it different? It will be helpful to have that at the beginning of the paper before methods.

In paragraphs four and five of the introduction we have provided more detail on how PTED and moral injury relate and differ from one another.

- Make sure to use the appropriate term. In conclusion, you mentioned "manager". Who are

they? Do you mean the healthcare authorities/ administration? Then explain that. A manager is not the right word!

We have replaced the word 'manager' with 'healthcare authorities'.

- This paper lacks a reasonable conclusion. Add more details to the conclusion to let it stand out.

At the end of the paper, we have now included a conclusion.

- I would highly recommend is to address the importance of Self-care along with organizational support. However, relying on administrative support only will bring more damage to mental health. Instead, some proactive self-care strategies should be implemented.

In the recommendation section of the discussion, we have now suggested that self-care strategies should focus on the significant predictors in this study i.e., increase self-esteem and reduce thinking about the consequences of one's actions.

-Overall, I think this is a strong paper. Again, it is the starting of many more empirical research on Covid-19.

Reviewer: 5

Dr. Charles W. Goss, Washington University in St Louis

Comments to the Author:

Thank you for the opportunity to review this paper that reports results from a survey of health and social care workers during the COVID-19 pandemic in the fall of 2020. This paper addresses moral and post-traumatic embitterment disorder (PTED) in response to the pandemic that is not widely reported in the literature. Overall, this is a well-written manuscript and a well-done study. I have some comments and suggestions aimed at improving the clarity and statistical reporting of the manuscript.

1. This study took place in the fall of 2020 and there was a proposed follow-up 1 year later. I can appreciate that a baseline paper such as this can be critical to get published prior to completion of the study, but it has been > 1 year since these data were collected and the manuscript indicates that longitudinal data should have been collected in the fall of 2021. Please provide an explanation for why longitudinal data were not included in this manuscript.

We agree that it would be beneficial to include follow up data, however this data has not yet been analysed and as you noted, this type of research is critical to get published as soon as possible.

2. The results subsection entitled “Preliminary results” is somewhat misleading as this oftentimes refers to non-definitive results from small studies (e.g., early-phase clinical trials). I think it could be removed or renamed more appropriately. Additionally, this section has very detailed descriptions of model diagnostics. While I appreciate the level of detail included, I think text could be made much more concise and some of the material could even be included in the methods.

We have removed preliminary results, and the information from this section has been condensed and combined with the ‘data analysis’ section from the method section.

3. Table 1: I would make separate tables for the descriptive statistics (mean, SD) and the Pearson correlation coefficient estimates.

Table 2 in the manuscript now includes the descriptive statistics that are separate from the correlation coefficient estimates.

4. Table 1: This table contains a lot of variables and is a bit challenging to interpret. I would recommend creating a correlation matrix heatmap. See this website for examples with various software packages: Heatmap Colored Correlation Matrix | LOST (lost-stats.github.io)

In line with your recommendation, we have included a correlation heatmap using R (see figure 1).

5. Page 12, lines 3-16: “Significantly” could be dropped altogether in the first sentence. As an example, “PTED was associated with older age ($P = XXX$) and existing self-reported mental health diagnoses ($P = XXX$)...”.

We have removed the word significantly from the univariate results section, and instead included the p values as part of the sentences in line with your recommendations.

6. There are several citations that are included in both the results and the methods (e.g., Cohen, Ferguson, Little). The citations are only needed in the methods.

These citations now only appear once in the method section.

7. Ferguson is misspelled in the methods and results as “Fergusson”. The year that this article was published is 2009 not 2016 as listed in the references.

We have included the correct spelling for Ferguson.

8. Specific significant (or not significant) variables in the results text should include associated P values.

Variables in the results text now all include their associated p values.

9. Rounding is inconsistent for percentages in the results. Sometimes there are 2 decimal places and sometimes 1. I would use a single decimal place.

We have used a single decimal place for all percentages.

10. Results should be formatted with a "0" before the decimal point if the value is < 1 (including P values).

The results have been formatted with "0" before the decimal point if the value is <1 now.

11. Page 6, line 37: Define "small to moderate effect size".

We have defined small to moderate effect size used in g power as $f^2 = .12$

12. Page 7, lines 20-21: The way this is phrased is confusing because the previous sentence references a factor analysis, but I believe that the "factor" in this sentence is not a factor score. Please clarify.

We have changed the word 'factor' to 'subscale'.

13. Page 11: remove "(two tailed)"

We removed (two tailed).

14. Page 23, lines 49-54: An important limitation is that only baseline (cross sectional) data were analyzed in this paper. This is clearly stated in the abstract, but it is not clear in the discussion.

We have now included the following sentence in the discussion, 'The most notable limitation is that cross-sectional baseline data was analysed, and only future analyses from our longitudinal study can establish causal relationships.'

15. The acronym NHS is not defined.

We have defined NHS in the introduction where it first appears now.

VERSION 2 – REVIEW

REVIEWER	Faisal Akram Saint Elizabeths Hospital/ DC Department of Behavioral Health, Psychiatry
REVIEW RETURNED	26-Mar-2022

GENERAL COMMENTS	Authors have adequately revised the manuscript; all my concerns have been addressed appropriately.
--

REVIEWER	Charles W. Goss Washington University in St Louis, Biostatistics
REVIEW RETURNED	01-Apr-2022

GENERAL COMMENTS	The authors have addressed all of my comments and I recommend that this manuscript should be accepted with the following minor revisions: 1. The statistics reported in the data-analysis methods should be in the results (e.g., test for MCAR). 2. Table 2: If +/- is for reporting the standard deviation, then parentheses are not needed, e.g., 36.84 +/- 10.73. 3. Table 2: where median IQR is reported, use: Median (Q1, Q3). 4. The correlation heat map looks great. The only suggestion that I would have is to remove the diagonal values (i.e., where correlations between the same variable = 1). This is more of a style/preference recommendation, and I will leave it at your discretion as how you want to present this.
--

VERSION 2 – AUTHOR RESPONSE

Reviewer: 5

Dr. Charles W. Goss, Washington University in St Louis

Comments to the Author:

The authors have addressed all of my comments and I recommend that this manuscript should be accepted with the following minor revisions:

1. The statistics reported in the data-analysis methods should be in the results (e.g., test for MCAR).

This information is now reported at the start of the results section instead.

2. Table 2: If +/- is for reporting the standard deviation, then parentheses are not needed, e.g., 36.84 +/- 10.73.

We have removed the parentheses.

3. Table 2: where median IQR is reported, use: Median (Q1, Q3).

We have included (Q1, Q3) for the IQR.

4. The correlation heat map looks great. The only suggestion that I would have is to remove the diagonal values (i.e., where correlations between the same variable = 1). This is more of a style/preference recommendation, and I will leave it at your discretion as how you want to present this.

Thanks. I have removed the diagonal values now and agree that it looks better.